# Physiological and Molecular Modulations to Drought Stress in the *Brassica* Species

**DOI:** 10.3390/ijms25063306

**Published:** 2024-03-14

**Authors:** Mi-Jeong Yoo, Yoojeong Hwang, Yoo-Min Koh, Fanchao Zhu, Aaditya Sunil Deshpande, Tyler Bechard, Silvana Andreescu

**Affiliations:** 1Department of Biology, Clarkson University, Potsdam, NY 13699, USA; hwangy@clarkson.edu; 2Duel Enrollment Program, University of Florida, Gainesville, FL 32610, USA; yoominkoh@ufl.edu; 3Morsani College of Medicine, University of South Florida, Tampa, FL 33612, USA; fanchaozhu@usf.edu; 4Department of Chemistry and Biomolecular Science, Clarkson University, Potsdam, NY 13699, USA; deshpaas@clarkson.edu (A.S.D.); becharta@clarkson.edu (T.B.); eandrees@clarkson.edu (S.A.)

**Keywords:** *Brassica*, polyploids, drought stress, superoxide, PFOS

## Abstract

Climate change, particularly drought stress, significantly impacts plant growth and development, necessitating the development of resilient crops. This study investigated physiological and molecular modulations to drought stress between diploid parent species and their polyploid progeny in the *Brassica* species. While no significant phenotypic differences were observed among the six species, drought stress reduced growth parameters by 2.4% and increased oxidative stress markers by 1.4-fold. Drought also triggered the expression of genes related to stress responses and led to the accumulation of specific metabolites. We also conducted the first study of perfluorooctane sulfonic acid (PFOS) levels in leaves as a drought indicator. Lower levels of PFOS accumulation were linked to plants taking in less water under drought conditions. Both diploid and polyploid species responded to drought stress similarly, but there was a wide range of variation in their responses. In particular, responses were less variable in polyploid species than in diploid species. This suggests that their additional genomic components acquired through polyploidy may improve their flexibility to modulate stress responses. Despite the hybrid vigor common in polyploid species, *Brassica* polyploids demonstrated intermediate responses to drought stress. Overall, this study lays the framework for future omics-level research, including transcriptome and proteomic studies, to deepen our understanding of drought tolerance mechanisms in *Brassica* species.

## 1. Introduction

Climate change brings about a variety of environmental shifts, including rising temperatures, altered precipitation patterns, shifting growing seasons, and an increase in pests and diseases due to warmer temperatures. One particularly significant consequence of these changes is the disruption of water cycles, which can have detrimental effects on plant growth and development. Notably, more frequent and intense droughts and floods have become increasingly common [1], resulting in reduced water availability for irrigation and subsequently leading to decreased crop yields. According to the National Centers for Environmental Information (NCEI), the economic impact of droughts and floods from 1980 to 2023 totaled USD 525 billion, with droughts alone accounting for USD 334.8 billion in damages [2]. These figures emphasize the urgent need to address the vulnerabilities of agriculture to the changing climate. As a response to these challenges, there has been an ongoing call for the development of crops resilient to climate change. To establish drought-tolerant crops, it is crucial to comprehend the mechanisms that underlie drought tolerance.

There have been many studies on drought responses in plants, including many crop species, e.g., wheat, maize, rice, canola, soybean, etc. [3,4,5,6,7]. These studies showed that plants have evolved a diverse range of morphological, physiological, and biochemical adaptations to endure drought stress effectively. Morpho-physiological traits related to drought responses include changes in plant height, leaf area, root system, relative water content (RWC), chlorophyll content, stomatal conductance, osmotic adjustments, and photosynthetic activities [8]. Decreased plant height, smaller leaf area, and thicker leaves were adapted traits for reduced photosynthetic capacity due to a water deficit [3,4,5,6,7,8]. Furthermore, plants exhibit reduced chlorophyll content and stomatal conductance due to the degradation of chlorophyll and stomatal closure during the day under drought conditions [3,4,5,6,7,8].

In contrast to their limited photosynthetic capacity, plants accumulate osmotic regulatory substances and drought-induced proteins, such as antioxidant enzymes. Osmoregulatory compounds are small molecules that are electrically neutral, easily dissolved, and do not harm cells, even at high concentrations. They play a big part in controlling abiotic stress in many ways. These compounds contribute to the maintenance of cell turgor pressure via osmoregulation, protect cell membranes, stabilize protein structures and other cellular components, and counteract the harmful effects of reactive oxygen species (ROS), thereby upholding cellular redox balance [9,10,11]. These osmoprotectants encompass amino acids (such as proline, γ-aminobutyric acid (GABA), citrulline, and pipecolic acid), betaines (e.g., glycine betaine), as well as sugars and sugar alcohols (including trehalose, sorbitol, inositol, and mannitol) [9,10,11].

Drought stress also induces molecular changes in plants, prompting the production of drought-responsive proteins such as antioxidants, aquaporin, cold shock proteins, cyclophilins, dehydrins, late embryogenesis abundant (LEA) proteins, heat shock proteins (HSPs), various transcription factors, molecular chaperones, and many enzymes and kinases that are involved in signaling pathways and hormone-mediated pathways [10,12]. Under drought conditions, ROS production can exceed the capacity of the ROS scavenging systems, which results in cell damage—such as lipid peroxidation, protein denaturation, DNA breakage, and blocked photosynthesis. Thus, maintaining ROS scavenging systems under stressed conditions is essential for stress tolerance in plants. Antioxidants play a role in protecting plants by scavenging ROS, which are antioxidant enzymes (such as superoxide dismutase (SOD), catalase (CAT), ascorbate peroxidase (APX), and peroxidase (POD)) and non-enzymatic antioxidants (including ascorbate, reduced glutathione (GSH), carotenoids, and flavonoids) [8,10].

The drought responses mentioned above have often been employed to assess drought stress in plants as well as develop drought-resistant crops, which are of polyploid origin. Polyploidy, also referred to as whole genome duplication (WGD), stands as a significant evolutionary mechanism prevalent across various eukaryotes. Notably, flowering plants exhibit the heightened tendency to experiencing WGD events [13,14,15,16,17,18,19]. Genome mergers have brought morphological, physiological, molecular, and biochemical novelty to polyploids, which often enable their better adaptability to harsh environments compared to diploid species [14,20,21,22,23,24,25,26,27,28,29,30]. This partially explains why most crops are of polyploid origin, which has been confirmed through genome sequencing [19,31,32]. Therefore, understanding how divergent genomes and their regulatory networks in polyploids are reconciled to respond to various environmental conditions is critical to elucidating the role of polyploidy in plant evolution and adaptation. In addition, it will offer a framework for the development of climate-resilient crops.

While drought response data for crops are available [3,4,5,6,7], there is still a gap in morphological and physiological information when comparing diploid parents with their corresponding polyploid species. For example, Barker et al. [33] investigated the relationships between leaf structure and function in both diploid and polyploid *Brassica* seedlings grown in controlled conditions. Their work showed that there was a significant correlation between morphological structure and physiological function in polyploids but not in diploids. This correlation was attributed to the expanded trait ranges resulting from the merging of genomes in polyploids [33]. Much research has indicated that polyploids exhibit higher drought tolerance compared to their diploid species; however, colchicine-induced autopolyploid species were investigated rather than natural polyploids and their diploid parent species [30,34,35,36,37]. Also, several studies examined diploids and polyploids together to see their responses to osmotic stress, but their work did not include both diploid parents and their polyploid species [38,39,40], except one study with *Brassica* [41], which showed higher salt tolerance in polyploids compared to diploid species.

Therefore, we aim to explore the morpho-physiological and molecular responses to drought stress in diploid species and their polyploid progeny. In this study, we focused on the *Brassica* system, also known as the U triangle [42], which contains three diploid species and three polyploid species derived from two diploid species (Figure 1). Although extant parental species are unknown or believed to be extinct in the wild, putative parental species are known for each polyploid species [43,44,45].

These six *Brassica* species are commercially significant crops such as vegetables and oilseeds. Furthermore, they are essential in the human diet due to their high nutrient content, antioxidant capabilities, potential cancer-preventive benefits, and contributions to different aspects of health [46,47,48]. Importantly, their genomes are fully sequenced [49,50,51,52,53,54], and some have been recently improved by the third-generation sequencing method [55,56,57,58,59]. In addition, all six *Brassica* species are well known for their osmotic stress tolerance, but polyploids outperform diploids in response to environmental stress [38,39,40,41] and have a higher biomass than their diploid parents [17]. Therefore, this system offers significant advantages in multi-omics studies in discerning the impact of polyploidy on plant speciation and evolution as well as facilitating the identification of molecular components crucial for improving stress-tolerant crops.

Here, we investigate morphological, physiological, and molecular differences in the six *Brassica* species in response to drought stress. We hypothesized that the increased genomic content in polyploids, compared to their diploid counterparts, enhances their adaptive capacity, leading to better drought tolerance than diploids. This study will lay the groundwork for future omics-level investigations, including transcriptomic and proteomic studies of the six *Brassica* species, focusing on identifying the common genetic components for drought tolerance. In particular, we performed targeted metabolite analysis of drought-responsive metabolites, such as abscisic acid (ABA), GABA, and three amino acids (phenylalanine, tryptophan, and proline). Most significantly, for the first time, we compared the amount of per- and polyfluoroalkyl substances (PFASs) between control and drought-stressed plants to evaluate this group of chemicals as indicators of drought stress in crops.

PFASs are synthetic compounds extensively utilized across various commercial and industrial sectors as surfactants in a diverse array of products, including repellent items such as leathers, fabrics, and foams [60,61]. Their widespread usage and their resistance to environmental breakdown, attributed to the strong carbon–fluorine bond in PFAS compounds, have led to the detection of numerous PFAS substances in the bloodstreams of people and animals worldwide. These compounds are present in trace amounts in various food products and throughout the environment [62]. Recent studies have shown that at least 45% of the nation’s tap water contains at least one type of PFAS [63], suggesting a broader and more concentrated distribution of these compounds than initially presumed. Given that plants absorb and store PFAS compounds in their leaves and roots [62,64,65], PFAS compounds have the potential to serve as indicators of drought stress since the reduced absorption of water during drought conditions may lead to decreased PFAS accumulation in plants.

## 2. Results

### 2.1. Morphological and Physiological Responses to Drought Stress

At the five-week-old stage, polyploid plants exhibit intermediate or transgressive phenotypes of their diploid parents in terms of leaf shape and size (Figure 2). After short-term drought stress, six *Brassica* species did not show dramatic differences in their phenotypes between the control and stressed groups (Figure 2). Some diploids exhibited white spots on leaves and rolled leaf margins under drought stress (Appendix A); however, they were excluded from further analysis as outliers.

Physiologically, however, drought-stressed plants showed decreased fresh weight, dry weight, and RWC compared to the control plants across all six species, although the differences in fresh and dry weights between control and stressed plants were not significant, except for *B. oleracea* and *B. carinata* (Appendix A). As for RWC, all six species exhibited a significant decrease of 2.4% on average in response to drought stress, except *B. napus* (Figure 3). Polyploid species exhibited RWC levels that were higher (*B. napus*), similar (*B. juncea*), or intermediate (*B. carinata*) compared to those of their diploid progenitors (Figure 3).

We also investigated ROS responses to drought stress by quantifying malondialdehyde (MDA), hydrogen peroxide (H_2_O_2_), and superoxide (O_2_^−^). As a result, an average increase of 1.4, 1.2, and 1.5-fold was observed in MDA, hydrogen peroxide, and superoxide levels in response to drought stress. While the MDA differences between control and stressed plants were significant, except in *B. rapa* and *B. juncea* (Figure 4A), only *B. oleracea* and *B. carinata* showed significant changes in hydrogen peroxide accumulation under drought conditions (Figure 4B). As for superoxide amounts, we observed significant changes across all six species when quantified using a biosensor (Figure 4C). The spectrophotometric measurements of superoxide showed an insignificant increase in the *Brassica* species in response to drought stress, except *B. rapa* (Figure 4D).

As drought stress leads to a reduction in chlorophyll contents, we also assessed the contents of chlorophyll a and b as well as carotenoids. As expected, the contents of each chlorophyll, total chlorophyll, and carotenoids decreased by an average of 13.2 to 14.6% in stressed plants, but the differences were not significant (Appendix A).

The principal component analysis (PCA) result showed the variation in morpho-physiological traits under control and drought conditions across all six species (Appendix A), but diploid species exhibited greater variation than polyploid species (Appendix A). Two factors, species and drought treatment, had significant effects on morpho-physiological traits, but their interactions had a substantial impact on the production of MDA, hydrogen peroxide, and superoxide (Appendix A).

### 2.2. Molecular Responses to Drought Stress

#### 2.2.1. Drought-Responsive Gene Expression

To examine the drought response at transcript levels, we surveyed eight genes known to be drought-responsive using quantitative real-time PCR (qRT-PCR), which include four genes encoding antioxidant enzymes, two early responsive to dehydration (*ERD*), and one late embryogenesis abundant (*LEA*) that are rapidly activated in response to drought stress. As for the genes responsible for ROS scavenging, three of them (iron superoxide dismutase (*FSD6*), manganese superoxide dismutase (*MSD2*), and catalase (*CAT3*)) showed increased expression in response to drought stress (Figure 5). Interestingly, polyploid species exhibited intermediate levels of expression compared to their diploid progenitors. However, the expression level of copper/zinc superoxide dismutase (*CSD1*) was decreased in drought-stressed plants compared to the controls, except *B. juncea* and *B. carinata*, which had similar expression levels under control and drought conditions (Figure 5). Among the four antioxidant genes, *FSD6* showed the highest expression changes in response to drought. While their expression levels under the control condition were similar across six species, other genes investigated here showed variation among the six species.

Regarding the expression changes in three *ERD* genes, the expression changes were variable across the species. For example, *B. oleracea* did not exhibit significant changes under drought conditions, while *B. rapa* and *B. juncea* showed higher expression in response to drought stress in all three *ERDs* (Figure 6). These *ERD* gene expression levels in polyploids were also intermediate or transgressive between two parental species, like antioxidant enzyme-encoding genes. The expression levels of *LEA4* were too low compared to other genes (e.g., 0.004 in *B. nigra*, 0.18 in *B. rapa* compared to *ACT1* under drought conditions), but this gene showed increased expression in response to drought stress across all six species (Figure 6).

The PCA result showed higher gene expression variation in *B. rapa* compared to other species for component 1 (*FSD6*, *CAT3*, *ERD1*, and *LEA4*; Appendix A), but no significant difference was observed. The expression of genes investigated here was affected by species and drought treatment as well as their interaction, except for *ERD1* (Appendix A).

#### 2.2.2. Drought-Responsive Metabolite Accumulation

Six drought-responsive metabolites and PFOS were measured from control and drought-stressed leaves. Levels of ABA, which trigger ABA-mediated signaling pathways in response to environmental stress and pathogen attacks, were increased, except in *B. carinata* (Table 1).

GABA, which rapidly accumulates in plants in response to abiotic and biotic stresses, also showed an increase except in *B. oleracea*. Three amino acids (phenylalanine, proline, and tryptophan) accumulated more in drought-stressed plants than in the control, but their increase was minimal or insignificant in some species (Table 1). Leucine, however, did not show changes under drought conditions, except in *B. nigra* and *B. napus*. Finally, PFOS, a long-chain PFAS, showed a decrease in all six species, as reflected in their reduced water uptake due to drought stress. The PCA results showed that GABA and ABA were more variable in diploid species than in polyploid species, while other metabolites were more variable across six species (Appendix A). An analysis of means for variances–Levene (ADM) showed that the levels of ABA and GABA varied more in diploid species compared to polyploid species (Appendix A). The abundance of metabolites investigated here were affected by either species, drought treatment, their interaction, or all three factors (Appendix A).

In addition, the absolute amount of PFOS was further quantified, which showed significant variation among six *Brassica* species. Polyploid species accumulated more PFOS compared to their diploid progenitors in control conditions (Table 2), partially in agreement with RWC data (Figure 3). Notably, *B. carinata* exhibited the highest level of PFOS accumulation in both conditions, while *B. oleracea* and *B. rapa* showed the lowest levels of PFOS accumulation in control and drought conditions, respectively.

## 3. Discussion

As sessile organisms, plants continuously interact with their ever-changing environments. Recent climate change, such as frequent and severe droughts and floods, poses detrimental effects on plant growth and development. Drought stress causes significant morphological, physiological, biochemical, and molecular changes in plants due to reduced water uptake. As a result, plants end their lifecycle earlier or invest more resources to tolerate drought stress, resulting in reduced productivity. Thus, understanding the mechanisms of drought response is critical to developing stress-tolerant crops. Considering the fact that crops are polyploids, a comparison of the responses to drought stress between diploid and polyploid species can shed light on how merged genomes can contribute to stress tolerance, which can be applied to the enhancement of crop quality and quantity under stressed conditions. For example, do duplicated individual genes increase stress tolerance, or is the duplicated gene network involved in stress tolerance? Although omics studies are useful for this purpose and prevalently practiced, morpho-physiological investigations should be carried out as preliminary works. Therefore, in this study, we employed the U triangle of *Brassica* species to examine morpho-physiological responses to drought stress. Additionally, we examined molecular features using qRT-PCR and metabolite profiling as 48 h stress might not be enough to bring about significant changes in morpho-physiology.

### 3.1. Wide Range of Variation in Response to Drought Stress across Six Species

Although there were no obvious morphological changes in response to drought stress (Figure 2), all six species exhibited reduced fresh weight, dry weight, and RWC (Figure 3 and Appendix A). Polyploid species were expected to have a higher biomass than diploid species due to hybrid vigor [66,67,68], which was not obvious here (Appendix A). Rather, polyploids exhibited intermediate values between their diploid parents, which were also observed in other polyploid species [69,70,71]. This might be explained by their early developmental stage, where there were no big differences in their plant body size or the number of leaves. Although polyploids show intermediate phenotypes, they exhibit differential responses to drought stress in other traits. For example, MDA, produced by membrane lipid peroxidation in response to ROS, was produced less in polyploids compared to diploids. *Brassica juncea* and *B. carinata* had less MDA than their diploid parents, while *B. napus* produced an intermediate amount of MDA between its diploid parents (Figure 4A). Interestingly, two diploids, *B. rapa* and *B. nigra*, produced relatively high amounts of MDA even in controlled conditions, and this can be explained by their developmental stress as their seedlings grew much faster than other *Brassica* species. These two species had relatively smaller seedlings after germination compared to the others, but they caught up with other species around five weeks. This feature was explained by PCA, which showed that these two diploids were separated from other species based on MDA and two ROS data (Appendix A). Two ROS molecules, hydrogen peroxide and superoxide, showed different results from each other. The amounts of hydrogen peroxide were similar across *Brassica* species, except *B. carinata*, which showed half the amount of hydrogen peroxide compared to other species (Figure 4B). All six species exhibited increased production of hydrogen peroxide in response to drought stress, but their changes were not significant due to high variability among different individual plants (Figure 4B). Other studies that examined hydrogen peroxide production under drought conditions showed a much higher increase compared to this study [39,72,73], but different developmental stages and treatments (e.g., drought duration) made direct comparisons unreliable. However, superoxide showed significant changes in response to drought stress across all six species (Figure 4C). Interestingly, *B. rapa* and *B. napus* seem to produce much higher amounts of superoxide in both conditions. The difference in accumulation of these two ROS molecules in *Brassica* species might be partially explained by the different techniques used in this study, which are further discussed below.

To see molecular responses to drought stress, the expression levels of several genes known as drought-responsive genes were investigated. As a result, three genes that encode the ROS-scavenging enzyme were highly expressed in all six species compared to the control conditions (Figure 5). There are three isoforms of *SOD* in plants, which are chloroplast and cytosolic *CSD*, chloroplastic *FSD*, and mitochondrial *MDS* [74,75]. In general, SOD activities were increased under stressed conditions, and their overexpression showed enhanced stress tolerance in many plants (reviewed in [74]). However, not all the plants investigated here exhibited an increase in *SOD* expression, suggesting there are considerable variations in SOD activity in response to abiotic stresses. There are also many different isoforms of *SOD* in *Brassica* species, 14–18 in *B. rapa*, 14 in *B. oleracea*, 31 in *B. napus*, and 29 in *B. juncea* [76,77]. *CSD* is the most common *SOD* in *Brassica*, like other plants, followed by *FSD* and *MSD*. The majority of *SOD*s responded to abiotic stresses, such as drought, salt, and cold stresses, but their responses were variable depending on the types of stress and developmental stages [76,77]. In this study, the expression of three *SOD* genes was investigated. *CSD1*, which is found in cells widely, exhibited lower expression in response to drought stress in all six species, while two organelle-specific *SOD*s (*FSD6*, *MSD2*) showed increased expression in all six species (Figure 5). The decreased expression of *CSD1* under drought conditions implies that some isoforms of *CSD* respond to other stressors or exhibit tissue-specific responses to stress. In *Arabidopsis*, *CSD1* exhibited the highest expression in stem tissue but lower expression or no change in response to drought stress (e.g., ABA, NaCl, and PEG treatments) [78]. Therefore, the lower expression of *CSD1* under drought conditions observed in this study might suggest the possibility of post-transcriptional regulations of *CSD1* expression or distinct responses of different *SOD* genes in response to stress. In addition to *SOD*, *CAT* expression was also investigated. After the conversion of superoxide to hydrogen peroxide by SOD, CAT metabolizes hydrogen peroxide to water and oxygen and reduces cellular ROS levels. There are three groups of *CAT*s which show tissue-specific and temporally variable expressions. *CAT1* is expressed in pollen and seeds, while *CAT2* and *CAT3* are mainly expressed in leaves, with the former in photosynthetically active tissues and the latter in vascular tissues and senescent leaves in *Arabidopsis* [79,80]. In this study, the expression of the *CAT3* homolog was examined, and a significant increase was observed only in *B. rapa* and *B. napus* (Figure 5). This result was partially supported by a previous study on *B. juncea* and *B. rapa*, which showed increased expression and activity of all CAT in response to drought stress [76]. Interestingly, similar responses were observed for *FSD6* and *CAT3* across six species in response to drought stress (Figure 5), indicating that these two may be working closely in peroxisomes where both FSD and CAT are commonly found. Although this seems reasonable based on their function, it should be further tested at the individual gene level and exploration in terms of their localization within cells. In addition to ROS-scavenging enzyme-encoding genes, three *ERD* genes were analyzed. *ERD* genes are known to quickly respond to drought stress and are expressed in the early stages of drought stress [81], but they are functionally diverse. ERD1 and ERD15 are transcription factors whose expression is induced by dehydration and high levels of salt [82], while ERD14, similar to ABA-induced class II LEA proteins, functions as a chaperone to protect several different enzymes [81]. In *Arabidopsis*, the accumulation of *ERD* enhanced tolerance to high levels of salt, drought, and low temperature [81]. In this study, three *ERD*s showed upregulation in response to drought stress, but their expression levels were variable across *Brassica* species, suggesting each species might respond to drought stress by regulating different types of *ERD* genes. As there are multiple *ERD* genes in diploid *Brassica* species (e.g., 14 in *B. oleracea*, 17–18 in *B. rapa* [83]), it would be meaningful to conduct a comparative analysis of the expression of *ERD* genes in all six *Brassica* species to see whether there are any commonly regulated *ERD* genes in response to drought stress.

When the expression changes in the genes investigated here were compared between diploid and polyploid species, significantly higher variations in the expressions were observed in diploid species (Appendix A). This result may imply that diploid species are more sensitive to drought stress compared to polyploids as they exhibit a less pronounced increase in gene expression. In fact, certain diploid plants displayed impaired phenotypes in response to drought stress, such as white spots on leaves and a rolled leaf margin (Appendix A), which have not been seen in polyploid species. The duplicated genomes in polyploid species might offer additional genomic components acquired by polyploidy, potentially enhancing their flexibility to modulate stress responses. However, the primers utilized in this study targeted both parental copies of genes, while the expression was normalized using an internal control gene, which is also duplicated in polyploids. Therefore, the interactions among duplicated genes via a duplicated regulatory network may offer a more comprehensive explanation for this result than focusing solely on the duplicated genes themselves. This hypothesis could be explored through genome-wide transcriptomic data analysis.

Several drought-responsive metabolites were also assessed using multiple reaction monitoring (MRM) to quantify them accurately. To cope with water stress, plants produce various metabolites that function as osmoprotectants to regulate the cellular water potential, such as amino acids, betaines, sugars, and sugar alcohols, which have been prevalently surveyed as drought-responsive metabolites. In this study, four amino acids (proline, leucine, tryptophan, and phenylalanine) and one hormone (ABA) were measured. Additionally, the abundances of GABA and PFOS were also determined, where the former is well known for responding to abiotic and biotic stresses, balancing the carbon/nitrogen ratio, and regulating plant development as a metabolite and a signal molecule [84,85,86,87,88], while the latter has not been measured in drought-stressed plants (refer to below). These metabolites showed increased abundance under drought conditions compared to the control (Table 1). Notably, two diploid species, *B. rapa* and *B. nigra*, exhibited stronger responses to drought stress than polyploid species, although there are some variations (Table 1). Of the seven metabolites studied, ABA, proline, and PFOS proved to be robust indicators of drought stress, showcasing significant changes across at least five species in response to drought stress (Table 1). In contrast to proline and ABA, which have been commonly studied in the context of drought stress, PFOS emerged as a novel drought marker in this study, being identified as such for the first time. Upon recognizing PFOS as merely one among several PFAS compounds, it is needed to explore the levels of other PFAS in plants to assess their potential as indicators of drought stress. In addition, further investigation into their tissue-specific accumulation patterns should be conducted.

### 3.2. Novel Approach in Assessing Drought Response

ROS, including hydrogen peroxide (H_2_O_2_), superoxide (O_2_^−^), singlet oxygen (^1^O_2_), and the hydroxyl radical (HO·), play an important role in plant development and growth [89,90,91,92]. These ROS molecules function as important signaling molecules, but they can also harm cells due to their toxicity. Stress-induced ROS accumulation can regulate stress-related signal transduction pathways, which in turn trigger gene expression involved in stress tolerance. In plants, two ROS molecules, hydrogen peroxide and superoxide, were commonly measured to evaluate the changes in ROS accumulation in response to stress, including drought and water stress. Hydrogen peroxide is the key signaling molecule in plants due to its relatively long half-life (on scales of seconds to minutes) and its cellular roles, such as oxidizing proteins and regulating protein function. In addition, the simple measurement method of hydrogen peroxide enabled this molecule to be used in stress monitoring in plants [93,94,95,96], as carried out in this study. In contrast, superoxide has an extremely short half-life (1–4 µs); thus, it cannot accumulate in cells. In general, either the nitroblue tetrazolium (NBT) staining method [97] or the spectrophotometric method [98] has been practiced to measure superoxide amounts. However, these methods produced variable data for ROS contents in plants due to their technical inaccuracies and sensitivity issues [99]. When we employed the spectrophotometric method, we also observed insignificant changes in the superoxide production rate in response to drought stress, except in *B. rapa* (Figure 4D). Thus, we employed a cytochrome C micro biosensor [100,101] to measure superoxide in plant extracts. This method enables real-time measurements and quantitative assessments of superoxide as it utilizes redox changes in cytochrome C, which was immobilized onto the gold electrode, in response to superoxides [100,101]. As a result, all six *Brassica* species showed a significant increase in superoxide amounts under drought conditions, suggesting the improved sensitivity of the biosensor method. As these two methods measure either the superoxide amount or superoxide production rate, a direct comparison between these two methods is not reliable. To ensure the reliability of the biosensor, it should undergo further testing with additional plant species across various developmental stages and under diverse stress conditions.

In this study, we assessed PFOS as a drought marker, as PFOS can trace water flow. As drought stress reduces water uptake, we hypothesized that drought-stressed plants contain smaller amounts of PFOS compared to well-watered plants. It is known that PFOA and PFOS are accumulated in leaves and roots, respectively [62,65,102], but PFOS concentrations were much higher than those of PFOA [103]. Thus, we examined PFOS due to its higher probability of being detected, and this study reports that PFOS accumulated less in drought-stressed plants than control plants (Table 1 and Table 2). The differences were significant in all six species, suggesting PFOS can be used as a drought marker. In addition, it is noteworthy that the accumulation of PFOS measured in this study is substantially higher compared to previous studies. For example, it was shown that a total amount of PFASs in plants from background soil was at pg/g (reviewed in [102]), while *B. oleracea* grown in fluorochemical manufacturing parks accumulated 3.38 ng·g^−1^ DW of PFOS [104]. In contrast, our study involved plants grown in uncontaminated soil and irrigated with tap water, both exhibiting negligible levels of PFOS (>0.1 ng/L). Therefore, our data suggest that the PFASs may accumulate at higher concentrations through irrigation than previously thought. However, further investigation is warranted to examine how accumulation patterns respond to varying input levels of PFASs. In addition, as mentioned earlier, other PFASs should be explored to see their potential as drought indicators. Finally, considering the adverse effect of PFAS on human health [105,106], further investigation should be carried into the accumulation of PFOS and other PFASs in other tissues, specifically roots and stems, which are targets of human consumption, along with leaf tissues of *Brassica*, such as radish, turnips, kohlrabi, and broccoli.

## 4. Materials and Methods

### 4.1. Plant Materials and Drought Stress Treatment

The seeds of six *Brassica* species were obtained from the Germplasm Resources Information Network (GRIN) of the U.S. Department of Agriculture (USDA) (Appendix A). Sterilized seeds were allowed to germinate on damp Whatman™ filter paper in Petri dishes at 23 °C for 3 days. After being pricked into the soil (Black Gold^®^ All-Purpose Potting Mix, Sun Gro Horticulture, Agawam, MA, USA), one-week-old seedlings with similar sizes were transplanted into a 2.37-L pot and grown in a growth chamber under a light intensity of 100–120 μmol m^−2^ s^−1^, 70–80% of relative humidity, and a 10 h light/14 h dark photoperiod at 23 °C for four more weeks by watering with 100 mL 0.5× Hoagland solution [107] every day. Then, five-week-old plants were treated with 25% polyethylene glycol (PEG) 6000 (water potential (ψ) = −1.032 MPa) to induce drought stress. After 48 h of treatment, one or two fully developed leaf tissues were collected from two independent plants per replicate, and five replicates were investigated.

### 4.2. Determination of Relative Water Content

Relative water content (RWC) in the leaves of control and treated seedlings was measured with the method of Smart and Bingham [108]. The fresh weight (FW) of leaves was recorded on the collection day, and they were floated in distilled water overnight, and the turgid weight (TW) was measured. Then, the same leaves were dried at 70 °C for 24 h to determine their dry weight (DW). RWC was calculated by the following Formula (1):RWC (%) = [(FW − DW)/(TW − DW)] × 100(1)

### 4.3. Photosynthetic Pigments Determination

To determine photosynthetic pigments, 500 mg of fresh leaf tissue was homogenized in 80% acetone. This homogenate was centrifuged at 3000× *g* for 20 min, and the optical density was spectrophotometrically recorded at 480, 646, and 663 nm for carotenoids, chlorophyll b, and chlorophyll a, respectively, with Thermo Scientific GENESYS 10S UV/visible scanning spectrophotometers (Thermo Fisher Scientific, Waltham, MA, USA) [109]. The contents of pigments were calculated according to Equations (2)–(5) [110,111,112].
*Cha* = 12.25 (A_663_) − 2.79 (A_646_)(2)
*Chb* = 21.5 (A_646_) − 5.1 (A_663_) (3)
*Total Chlorophyll* = (5.24 (A_663_) + 22.24 (A_646_))/FW(4)
*Carotenoid* = (1000 (A_470_) − 1.82 *Cha* − 85.02 *Chb*)/198 (5)

### 4.4. Measurement of MDA

MDA is a substance produced by membrane lipid peroxidation in response to ROS [113,114]. For the quantification of MDA content, we followed the method of Heath and Packer [115]. First, 100 mg of fresh leaf tissue was macerated in a 1% trichloroacetic acid (TCA) solution and then centrifuged at 10,000× *g* for 10 min. Then, 1 mL of supernatant was mixed with 1 mL of 0.5% thiobarbituric acid and boiled at 95 °C for 30 min. After cooling the tubes in an ice bath, they were centrifuged at 5000× *g* for 5 min, and the supernatant was extracted to record the optical density at 532 and 600 nm. The concentration of MDA was calculated using the following Formula (6):MDA equivalents (nmol/mL) = [(A_532_ − A_600_)/155,000] × 106 (6)
where 155,000 is the molar extinction coefficient for MDA.

### 4.5. Measurement of Reactive Oxygen Species Content (Hydrogen Peroxide, Superoxide)

To determine the levels of hydrogen peroxide, 200 mg of fresh leaf sample was homogenized in 0.1% TCA and centrifuged at 12,000× *g* for 15 min. Then, 500 µL of supernatant was mixed with 500 µL of 10 mM potassium phosphate buffer (pH 7.0), followed by the addition of 1 mL of 1 M potassium iodide. Finally, the optical density of the sample was recorded at 240 nm [116].

The content of superoxide was determined by using the spectrophotometric method [98] and an implantable Cytochrome C micro biosensor [100,101]. In the spectrophotometric method, we followed Elstner and Heupel’s study [98], which recorded absorbance at 530 nm and used a standard curve with NaNO_2_ to calculate the superoxide production rate. The superoxide production rate was expressed as nmol·min^−1^·g^−1^ on a fresh weight basis. In the measurement with a biosensor, 200 mg of fresh leaf sample was homogenized with 1 mL of PBS (pH 6.5) buffer and centrifuged at 12,000× *g* for 15 min at 4 °C. Then, 200 µL of the supernatant was taken and loaded into an electrochemical cell containing 1% TCA made in a 0.1 M PBS solution, making a total volume of 4.2 mL. An increase in the current was noticed as soon as the sample was added, and the resultant increase in the current was subtracted from the baseline to yield the actual value. The concentration of superoxide was calculated using the calibration equation. Prior to this experiment, the sensor was calibrated in the TCA solution for superoxides, which yielded a sensitivity of 6.79 µM/nA and y = 6.79x + 0.6419 as the calibration equation.

### 4.6. Assay of Expression of Drought-Responsive Genes Using Quantitative Real-Time PCR (qRT-PCR)

To measure the expression levels of genes that respond to drought stress, the qRT-PCR method was employed. Based on the previous studies, we selected eight genes, including four ROS-scavenging enzyme-coding genes (*CAT*, *CSD*, *FSD*, *MSD*), three *ERDs*, and one *LEA*, and the primers were designed to target common sequences of genes from six species (Appendix A). RNAs were extracted using the Qiagen RNeasy Plant mini kit (Qiagen, Stanford, CA, USA) and quantified using NanoDrop™ One spectrophotometers (Thermo Fisher Scientific, Waltham, MA, USA). After the removal of DNA using DNase I (New England Biolabs, Ipswich, MA, USA), cDNAs were synthesized with the cDNA synthesis kit (New England Biolabs, Ipswich, MA, USA). For qRT-PCR, Luna Universal qPCR Master Mix (New England Biolabs, Ipswich, MA, USA) was used with CFX90 (Bio-Rad, Hercules, CA, USA) following the manufacturer’s protocol. For each reaction, two technical and three biological replicates were included. The relative expression of the target genes was calculated using the comparative Ct method (Applied Biosystems, Framingham, MA, USA), and *ACT1* was used as an internal control.

### 4.7. Measurement of Metabolites Using MRM

Seven metabolites (ABA, GABA, four amino acids, and PFOS) were measured in the control and drought-stressed samples. We focused on PFOS within the PFAS group due to its widespread prevalence and greater tendency for bioaccumulation due to its long chain [117,118]. Analytical standards, including γ-aminobutyric acid (GABA; Cat. No. 03835-250MG; analytical standard), 4-aminobutyric acid-2,2-d_2_(GABA-d_2_, Cat. No. 617458-250MG; purity > 98 atom % D), DL-proline (Cat. No. 171824-1G; purity > 99%), L-leucine (Cat. No. L8000-25G; purity > 98%), L-tryptophan (Cat. No. T0254-1G; purity > 98%), L-phenylalanine (Cat. No. P2126-100G; purity > 98%), (+)-abscisic acid (Cat. No. 90769-25MG; analytical standard), perfluorooctanesulfonic acid (PFOS; Cat. No. 77283-10ML; analytical standard), (1R)-(−)-10-camphorsulfonic acid (CA; Cat. No. 282146-25G; purity > 98%), were purchased from Sigma Chemical Co. (St. Louis, MO, USA), while perfluorooctanesulfonate [^13^C_8_]-Na salt (PFOS) (537.1 Internal Primary Dilution Standard; Cat. No. PFS-537-IPDS) was obtained from Agilent (Santa Clara, CA, USA).

#### 4.7.1. Sample Preparation

The lyophilized samples (10 mg dry weight) were homogenized with a ceramic bead using the TissueLyser II instrument (Qiagen, Hilden, Germany) set at 30 strokes/s for 2 min. The metabolites were extracted following an established method [119]. Briefly, the samples were extracted once in 1 mL of extraction solvent I (acetonitrile/isopropanol/water, 3:3:2), in addition to 10 µL of camphorsulfornic acid in methanol (10 ng/mL), 10 µL of GABA-d_2_ (10 ng/mL), and 10 µL of PFOS-^13^C_8_ (10 ng/mL) and twice with 0.5 mL of extraction solvent II (acetonitrile/water, 1:1) on a thermomixer (Thermomixer R, Eppendorf, Hamburg, Germany), mixing at a speed of 1200 rpm at 4 °C for 10 min each time, followed by sonication for 15 min on ice and centrifugation at 15,000× *g* for 15 min at 4 °C. Prior to extraction, the internal standard mixture with 2 µL of 100 nM of 10-camphorsulfonic acid, 10 µL of GABA-d_2_ (100 ng/mL), and 10 µL of PFOS-^13^C_8_ (100 ng/mL) was added to each sample. The supernatants were combined and lyophilized, and the extracted compounds were dissolved in 50% methanol (*v*/*v*) and spun down at 16,000× *g* for 30 min for LC-MS analysis. The injection volume was 1 µL.

#### 4.7.2. Optimization and Quantitation Using LC-MSMS

The compounds were analyzed using the triple quadrupole TSQ Altis mass spectrometers (Thermo Fisher Scientific, San Jose, CA, USA) coupled with the Vanquish Horizon UHPLC (Thermo Fisher Scientific). The TSQ Altis was housed with a heated electrospray ionization (HESI) source using the following source settings: sheath gas flow of 50 Arb, auxiliary gas flow of 10 Arb, sweep gas flow of 1 Arb, AND ion transfer tube temperature of 325 °C. The vaporizer temperature was set at 350 °C, and the spray voltage was 4.0 kV under the positive polarity and 2.5 kV under the negative polarity. The scan time was set at 1 s, and the Q1 and Q3 resolutions of full width at half maximum (FWHM) were both 0.7. For the CID gas pressure, 1.5 mTorr was used. To determine the optimal fragments and collision energies (CES) for MRM transitions, we utilized the Xcalibur 4.1 software from Thermo Fisher Scientific for optimization. The selected fragment ions of each precursor with collision energies, including isotopically labeled precursors, are described in Table 3.

To quantify the metabolites, each sample was loaded and separated on a Zorbax RRHD Eclipse Plus C18 analytical column (50 mm × 2.1 mm; 1.8 μm particle) at a flow rate of 0.4 mL/min. For the positive mode, 0.1% formic acid (*v*/*v*) was used in both solvent A (5 mM ammonium acetate) and B (100% acetonitrile). Linear gradients were set up from solvent A to 1% solvent B for 1 min, to 20% solvent B for 3 min, to 40% solvent B for 4 min, ramping up to 98% solvent B for 4.8 min, holding it for 1.2 min, ramping up to 99% solvent A for 0.5 min, and holding it for an additional 1.5 min. As for PFOS, the instrumental analysis followed US Environmental Protection Agency (EPA) Methods 537.1 [118] and 8327 [119]. Briefly, the Vanquish Horizon UHPLC system was installed with the PFAS Analysis Kit (Cat. No. 80000-97020, San Jose, CA, USA), and an Hypersil Gold C18 (50 mm × 4.6 mm; 1.9 µm particle, Thermo Scientific, San Jose, CA, USA) was used as the isolator column. To detect PFOSs in plants, the samples were loaded and separated on an Accucore RP-MS analytical column (100 × 2.1 mm, 2.6 µm particle, Thermo Scientific) at a flow rate of 0.4 mL/min. Five mM ammonium acetate in Optima LC-MS grade water (Thermo Scientific) was used in solvent A, while Optima LC-MS grade acetonitrile was used in solvent B. A linear gradient was set up from solvent A to 30% solvent B for 1.5 min, ramping up to 100% solvent B for 7 min, holding it for 4.5 min, ramping 100% solvent A for 1.5 min, and holding it for an additional 2 min.

#### 4.7.3. Data Process of Metabolites

Xcalibur 4.1 software (Thermo Fisher Scientific) was used to determine the compounds, and the Quant Brower was employed in quantification. To obtain the absolute expression value of the metabolite, the peak area of the compound was normalized by the initial amounts of samples and the input amount of the isotopic GABA-d_2_, PFOS-^13^C_8_, and CA as the internal standards in positive and negative modes, respectively. This normalized peak area was then compared to the standard curve of the metabolite standards, allowing for the determination of its quantitative value.

### 4.8. Data Analysis

Differences between control and drought-stressed samples in each *Brassica* species were evaluated using Student’s *t*-test. Differences were considered significant if the *p*-value was < 0.05 for morpho-physiological data and the *p*-value was < 0.1 for gene transcript and metabolite data. All the data were presented as mean ± standard error. Correlations among the measured values and responses of the *Brassica* species were explored via PCA-, which was performed using the correlation matrix of the average values of traits after standardization (autoscaling) with JMP Pro 15.0.0 (SAS Institute Inc., Cary, NC, USA). Also, the effects of species and treatment were examined via a two-way ANOVA. Finally, an analysis of means for variances–Levene (ADM) was conducted using JMP Pro 15.0.0 to see if there was variation among the samples.

## 5. Conclusions

This study investigated morpho-physiological responses, along with molecular and metabolite profiles, in response to drought stress in six *Brassica* species. Three diploid species and their polyploid progenies were compared in terms of drought responses, but there were no significant phenotypic differences. However, drought stress resulted in decreased morpho-physiological features, such as fresh and dry weights, RWC, and pigment contents while increasing the ROS-related molecules like MDA, hydrogen peroxide, and superoxide. Despite similar responses across six *Brassica* species, diploid species exhibited more variation compared to their polyploid progenies. Similar trends were observed for molecular responses. Transcripts of genes encoding ROS-scavenging enzymes and known to be drought-responsive were more abundant in drought-stressed plants relative to the control plants. Targeted metabolite profiling also showed their accumulation in response to drought stress, except PFOS, which showed reduced accumulation. In general, all six species exhibited a wide range of variation in morpho-physiological, molecular, and metabolite features, while polyploid species showed less variation compared to their diploid parent species. This study compared all six species under the same conditions, showing that both species and treatment played a significant role in each species’ responses, along with the interaction between species and treatment. However, their very similar responses to drought stress will provide a framework for transcriptomic or proteomic profiles in these six *Brassica* species as this system will be useful for identifying either shared or unique transcriptomic networks or proteomic regulations in response to drought stress.

## Figures and Tables

**Figure 1 ijms-25-03306-f001:**
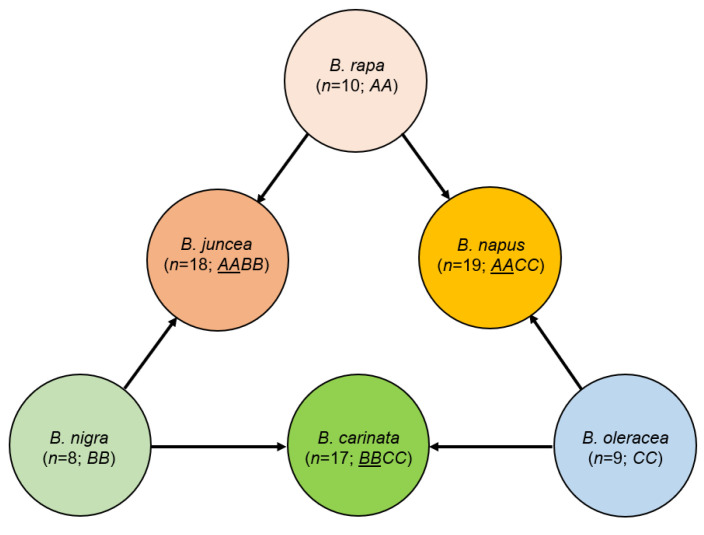
The triangle of U [42]. Underlined letters in parentheses represent the putative maternal parental genome in polyploids.

**Figure 2 ijms-25-03306-f002:**
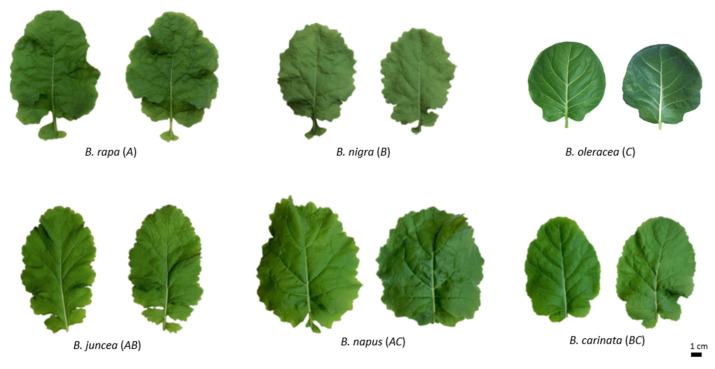
Morphological comparison between control and drought-stressed leaves in *Brassica* species. In each comparison, the left and right ones represent control and drought-stressed leaves, respectively.

**Figure 3 ijms-25-03306-f003:**
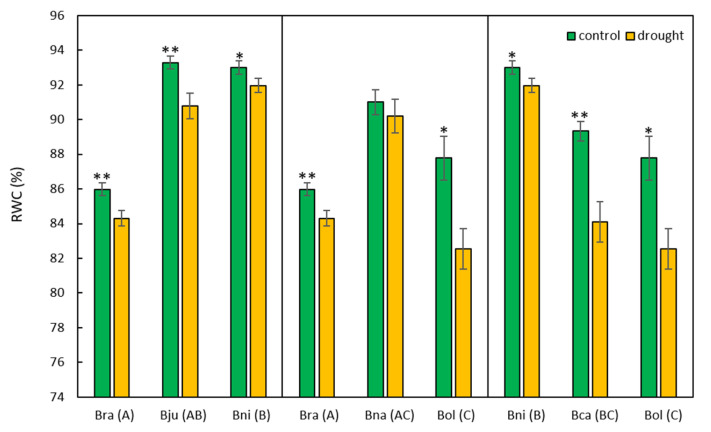
Comparison of RWC between control and drought-stressed plants of *Brassica* species. Bra (A) = *B. rapa* (A), Bju (AB) = *B. juncea* (AB), Bni (B) = *B. nigra* (B), Bna (AC) = *B. napus* (AC), Bol (C) = *B. oleracea* (C), Bca (BC) = *B. carinata* (BC). Diploid species data were presented twice to compare the values of polyploids relative to those of their diploid parents. Data were presented as mean ± standard error. * = *p*-value < 0.05, ** = *p*-value < 0.01.

**Figure 4 ijms-25-03306-f004:**
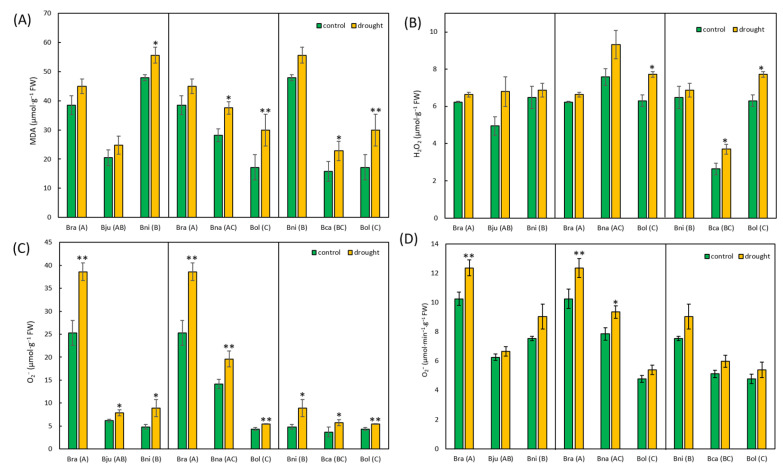
Comparison of MDA (**A**), hydrogen peroxide (H_2_O_2_), (**B**) superoxide (O_2_^−^) measured with a biosensor (**C**) and spectrophotometric method (**D**) between control and drought-stressed plants of *Brassica* species. The averaged fold change was presented as a comparison among species (**D**). Data were presented as mean ± standard error. * = *p*-value < 0.05, ** = *p*-value < 0.01.

**Figure 5 ijms-25-03306-f005:**
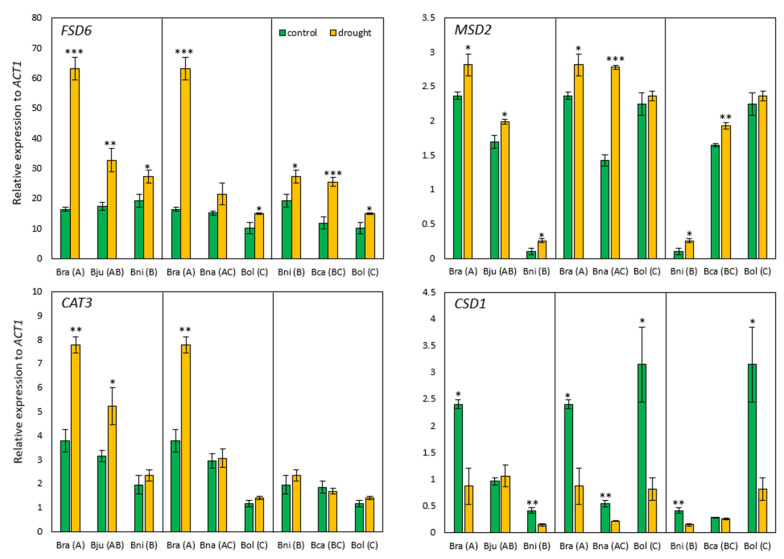
qRT-PCR results for four genes involved in ROS scavenging. The expression of each gene was normalized to an internal control *actin* gene (*ACT1*), and green and orange bars represent control and drought-stressed plants, respectively. Data were presented as mean ± standard error. * = *p*-value < 0.1, ** = *p*-value < 0.05, *** = *p*-value < 0.01.

**Figure 6 ijms-25-03306-f006:**
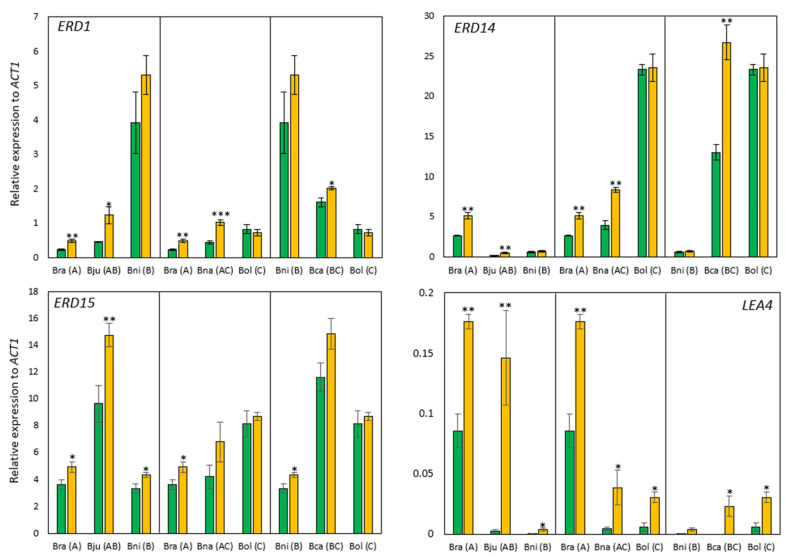
qRT-PCR results of four genes known to be drought-responsive. The expression of each gene was normalized to an internal control *actin* gene (*ACT1*), and green and orange bars represent control and drought-stressed plants, respectively. Data were presented as mean ± standard error. * = *p*-value < 0.1, ** = *p*-value < 0.05, *** = *p*-value < 0.01.

**Table 1 ijms-25-03306-t001:** The profiles of six drought-responsive metabolites. Log2 fold change was calculated as the ratio between drought-stressed plants and control plants, using control as the denominator. * = *p*-value < 0.1, ** = *p*-value < 0.05, *** = *p*-value < 0.001, n/s = not significant.

Species	ABA	GABA	Leucine	Tryptophan	Phenylalanine	Proline	PFOS
*B. rapa*	4.83 **	1.14 **	0.92 ^n/s^	9.66 **	4.70 **	2.42 ***	−4.08 **
*B. nigra*	6.14 **	0.57 *	2.00 **	2.58 **	2.27 **	1.86 **	−0.42 *
*B. oleracea*	1.63 ***	1.09 ^n/s^	0.81 ^n/s^	0.21 **	−0.12 ^n/s^	0.59 **	−0.28 **
*B. juncea*	0.31 *	0.81 **	0.91 ^n/s^	1.48 **	0.27 ^n/s^	0.0057 ^n/s^	−1.24 **
*B. napus*	0.39 **	0.76 **	1.25 **	0.40 ^n/s^	0.61 **	0.76 **	−2.14 *
*B. carinata*	0.03 ^n/s^	0.33 **	0.96 ^n/s^	0.24 ^n/s^	0.13 ^n/s^	0.63 **	−0.25 **

**Table 2 ijms-25-03306-t002:** The amounts of PFOS measures in the *Brassica* species. Mean ± standard error.

Species	Control (ng·g^−1^ DW)	Drought (ng·g^−1^ DW)
*B. rapa*	3.735 ± 0.079	0.220 ± 0.002
*B. nigra*	4.281 ± 0.248	3.209 ± 0.032
*B. oleracea*	2.046 ± 0.056	1.688 ± 0.052
*B. juncea*	4.470 ± 0.050	1.889 ± 0.123
*B. napus*	3.759 ± 0.235	0.854 ± 0.028
*B. carinata*	5.944 ± 0.037	4.985 ± 0.076

**Table 3 ijms-25-03306-t003:** Summary of precursor-to-product ion transitions used for the quantification of metabolites using RP-LC/ESI–MS/MS.

Compound	Polarity	Precursor (*m*/*z*)	Product (*m*/*z*)	Collision Energy (V)	Min Dwell Time (ms)	RF Lens (V)	Use Quan Ion
GABA	Positive	104	45	21	40	30	FALSE
GABA	Positive	104	87	13	40	30	TRUE
GABA-d_2_	Positive	106	69.92	21	40	30	FALSE
GABA-d_2_	Positive	106	88.92	13	40	30	TRUE
DL-proline	Positive	116.1	43	29	40	30	FALSE
DL-proline	Positive	116.1	70	16	40	30	TRUE
L-leucine	Positive	132.1	86	11	40	30	TRUE
L-leucine	Positive	132.1	44	24	40	30	FALSE
L-tryptophan	Positive	205.1	146	20	40	40	FALSE
L-tryptophan	Positive	205.1	188	10	40	40	TRUE
L-phenylalanine	Positive	166.1	120	15	40	31	TRUE
L-phenylalanine	Positive	166.1	103	28	40	31	FALSE
CA	Negative	231.1	79.9	32	50	76	TRUE
CA	Negative	231.1	149.1	25	50	76	FALSE
(+)-Abscisic acid	Negative	263.3	153	11	50	48	TRUE
(+)-Abscisic acid	Negative	263.3	219	13	50	48	FALSE
PFOS	Negative	499	79.9	43	40	200	TRUE
PFOS	Negative	499	98.9	41	40	200	FALSE
PFOS-^13^C_8_	Negative	507	79	43	40	208	FALSE
PFOS-^13^C_8_	Negative	507	99	39	40	208	TRUE

## Data Availability

Data is contained within the article or Appendix A.

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
