# Peer review of "Physiological and Molecular Modulations to Drought Stress in the Brassica Species"

_ijms, 2024, doi:10.3390/ijms25063306_

Round 1

Reviewer 1 Report

Comments and Suggestions for Authors

Title: Physiological and molecular modulations to drought stress in the Brassica species

Resubmit after significant improvements

The present study focuses on Physiological and molecular modulations to drought stress in the Brassica species. This study project is intriguing, creative, and technically sound. Furthermore, the

experimental data supports the authors& claim, and the statistical analysis was sufficient and

reliable. However, the writing of the manuscript needs to be improved before it is considered

for publication.

Abstract

1. Line 5: The growth attributes were reduced by how much percent, and reactive oxygen species increased? Rewrite the results of abstract in percentage.

2. Line 12: Clearly mention the genomic components.

3. Title of research article: "Physiological and Molecular Modulation" should be changed to "Physiological and Molecular Modulation" in the abstract.

4. Abstract needs much improvement.

   Introduction

5. Line 53-56: Reference is missing.

6. Line 66-68: Reference is missing.

7. Add hypothesis of your study.

   Results

8. Add results in percentage.

9. Add lettering in Table 2 with mean and standard error.

10. In results, you have mentioned (Figure S1, S2, S3) as Figure 1, 2, 3. These figure numbers are confusing in the whole manuscript.

   Material and methods

11. Line 470: How is it possible that after 48 hours of imposition of drought, growth attributes are reduced?

12. Line 480: Photosynthetic attributes are missing in the table and figure.

Comments on the Quality of English Language

-

Author Response

We appreciate you and the reviewers’ precious time in reviewing our paper and providing valuable comments. We have carefully and thoughtfully revised the manuscript and incorporated the suggestions of the reviewers insofar as they seem reasonable and helpful. Detailed responses to each itemized comment from the three respective reviewers follow. Additionally, a fluent English user has reviewed this manuscript. All modifications in the manuscript have been highlighted in red.

Response to Reviewer 1

  1. Line 5: The growth attributes were reduced by how much percent, and reactive oxygen species increased? Rewrite the results of abstract in percentage.

We incorporated your comments in abstract as below [lines 17-19].

While no significant phenotypic differences were observed among the six species, drought stress reduced growth parameters by 2.4% and increased oxidative stress markers by 1.4 times.

  1. Line 12: Clearly mention the genomic components.

We hypothesize that the increased genomic content in polyploids compared to their diploid counterparts may influence stress tolerance. For instance, duplicated stress-responsive genes or their regulatory networks could enhance their adaptive capacity. However, comprehensive genome-wide studies are needed to confirm this hypothesis. Therefore, we propose that the duplicated genomes likely play a significant role in bolstering stress tolerance in polyploids. To clarify this point, we revised the abstract below [line 26].

“their additional genomic components acquired through polyploidy may improve their flexibility to modulate stress responses.”

  1. Title of research article: "Physiological and Molecular Modulation" should be changed to "Physiological and Molecular Modulation" in the abstract.

We changed “morpho-physiological and molecular responses” to “physiological and molecular modulation” in the abstract [lines 15-16].

  1. Abstract needs much improvement.

We appreciate your comments and think that our revisions based on your feedback have improved the clarity of the abstract.

  1. Line 53-56, Line 66-68: Reference is missing.

Proper references are cited.

  1. Add hypothesis of your study.

We added hypothesis in Lines 137 – 139.

“We hypothesized that the increased genomic content in polyploids, compared to their diploid counterparts, enhances their adaptive capacity, leading to better drought tolerance than diploids..”

  1. Add results in percentage.

We added percentages or fold-change in results [Lines 178, 192, 211].

  1. Add lettering in Table 2 with mean and standard error.

As the log2 Fold change was derived from the averaged values of replicates, it is not possible to determine the mean and standard error. The terms "mean and standard error" were mistakenly included in the table legend and have been removed.

  1. In results, you have mentioned (Figure S1, S2, S3) as Figure 1, 2, 3. These figure numbers are confusing in the whole manuscript.

We attempted to distinguish supplementary figures from main figures by labeling them as "supplementary Figure S1". We carefully examined the figures we mentioned to eliminate any confusion.

  1. Line 470: How is it possible that after 48 hours of imposition of drought, growth attributes are reduced?

We established the length of drought stress by referring to prior research that utilized PEG 6000. Various studies indicated that subjecting plants to this treatment for 24 to 48 hours led to decrease in chlorophyll levels and increase in antioxidant enzyme activity across different plant species, including sorghum (Jung et al., 2015), tobacco (Yang et al., 2017), Arabidopsis (Fu et al., 2018), and peanut (Shivakrishna et al., 2018). Although we used 48-hour of treatment, we have a plan to investigate the morpho-physiological features in a time-course in the future.

  • Fu, Y., Ma, H., Chen, S., Gu, T., & Gong, J. (2018). Control of proline accumulation under drought via a novel pathway comprising the histone methylase CAU1 and the transcription factor ANAC055. Journal of Experimental Botany, 69(3), 579-588. https://doi.org/10.1093/jxb/erx419
  • Jung et al. 2015. Effects of Polyethylene Glycol-Induced Water Stress on the Physiological and Biochemical Responses of Different Sorghum Genotypes. The XXIII International Grassland Congress (Sustainable use of Grassland Resources for Forage Production, Biodiversity and Environmental Protection). New Delhi, India.
  • Shivakrishna, P., Reddy, K. A., & Rao, D. M. (2018). Effect of PEG-6000 imposed drought stress on RNA content, relative water content (RWC), and chlorophyll content in peanut leaves and roots. Saudi Journal of Biological Sciences, 25(2), 285-289. https://doi.org/10.1016/j.sjbs.2017.04.008
  • Yang, H., Zhao, L., Zhao, S., Wang, J., & Shi, H. (2017). Biochemical and transcriptomic analyses of drought stress responses of LY1306 tobacco strain. Scientific Reports, 7(1), 1-10. https://doi.org/10.1038/s41598-017-17045-2

  1. Line 480: Photosynthetic attributes are missing in the table and figure.

Line 180 does not include information on photosynthetic attributes, but we assessed the chlorophyll levels for photosynthetic attributes, which were presented in supplemental Figure S3.

Reviewer 2 Report

Comments and Suggestions for Authors

Dear author's,

all comments are inserted into the file.

I found your paper interesting, but you should have in mind that seedlings are mostly produced in controled environment (greenhouses, etc.) with regular watering and fertigation, with very low possibility of drough or temperature stress. My opinion is that your results would be more representative if you plant seedligs of tested species at the open field under different watering regimes.

Author Response

We appreciate you and the reviewers’ precious time in reviewing our paper and providing valuable comments. We have carefully and thoughtfully revised the manuscript and incorporated the suggestions of the reviewers insofar as they seem reasonable and helpful. Detailed responses to each itemized comment from the three respective reviewers follow. Additionally, a fluent English user has reviewed this manuscript. All modifications in the manuscript have been highlighted in red.

Response to Reviewer 2

  1. all comments are inserted into the file.

The manuscript was revised as suggested.

  1. I found your paper interesting, but you should have in mind that seedlings are mostly produced in controlled environment (greenhouses, etc.) with regular watering and fertigation, with very low possibility of drought or temperature stress. My opinion is that your results would be more representative if you plant seedlings of tested species at the open field under different watering regimes.

 Thank you for your recommendations. As we studied six species in two different conditions, we chose to employ a controlled environment for easier handling and interpretation of results compared to a field investigation with more variables to consider. We plan to validate our results through fieldwork in the future.

Reviewer 3 Report

Comments and Suggestions for Authors

The growth conditions must be much better described:  What temperature, light, CO2 and humidity conditions occurred during growth.  Field, indoor chambers or glasshouse?  Type of soil?  Water potential of the PEG?

I did not see statistical tests of differences in variability as were claimed.  If this was the main finding, it needs to be statistically convincing. 

Author Response

We appreciate you and the reviewers’ precious time in reviewing our paper and providing valuable comments. We have carefully and thoughtfully revised the manuscript and incorporated the suggestions of the reviewers insofar as they seem reasonable and helpful. Detailed responses to each itemized comment from the three respective reviewers follow. Additionally, a fluent English user has reviewed this manuscript. All modifications in the manuscript have been highlighted in red.

Reponses to Reviewer 3

  1. The growth conditions must be much better described:  What temperature, light, CO2 and humidity conditions occurred during growth.  Field, indoor chambers or glasshouse?  Type of soil?  Water potential of the PEG?

Thank you for your comments. We revised the methods by providing the growth condition, soil, and water potential of the PEG in lines 514-521. We assume that CO2 would be around 425 ppm as the plants were grown in normal air condition, so it is not presented in the manuscript.

  1. I did not see statistical tests of differences in variability as were claimed.  If this was the main finding, it needs to be statistically convincing. 

We included statistical deviation to show higher variability in diploid species compared to polyploidy species in Lines 689-690. This comparison was included in Supplemental Information as Supplementary Figure 5S.

Round 2

Reviewer 3 Report

Comments and Suggestions for Authors

The issues that I raised have been adequately addressed.